# Effect of Spin Coating Parameters on the Electrochemical Properties of Ruthenium Oxide Thin Films

**Elisabetta Petrucci** [1], **Monica Orsini** [2], **Francesco Porcelli** [2], **Serena De Santis** [2] and **Giovanni Sotgiu** [2,*]

1   Department of Chemical Engineering Materials & Environment, Sapienza University of Rome, 00184 Rome, Italy; elisabetta.petrucci@uniroma1.it
2   Department of Engineering, Roma Tre University, 00146 Rome, Italy; monica.orsini@uniroma3.it (M.O.); francesco.porcelli@uniroma3.it (F.P.); serena.desantis@uniroma3.it (S.D.S.)
*   Correspondence: giovanni.sotgiu@uniroma3.it

**Abstract:** Ruthenium oxide (RuOx) thin films were spin coated by thermal decomposition of alcoholic solutions of $RuCl_3$ on titanium foils and subsequently annealed at 400 °C. The effect of spin coating parameters, such as spinning speed, volume, and molar concentration of the precursor as well as the number of deposits, on the morphology and electrochemical performance of the electrodes was investigated. The films were characterized by scanning electron microscopy (SEM) equipped with energy-dispersive X-ray spectroscopy (EDX), cyclic voltammetry (CV) with and without chloride, and linear sweep voltammetry (LSV). The prepared materials were also compared to drop cast films and spin-coated films obtained by adopting low-temperature intermediate treatments. The results indicate that even dispersion of the oxide layer was always achieved. By tuning the spin coating parameters, it was possible to obtain different electrochemical responses. The most influential parameter is the number of deposits, while the concentration of the precursor salt and the rotation speed were less relevant, under the adopted conditions.

**Keywords:** ruthenium oxide electrode; thin film; spin coating; voltametric charge; linear sweep voltammetry; chloride effect

## 1. Introduction

Ruthenium oxide-based electrodes have long been considered extremely attractive materials, mainly because of their high conductivity, easy preparation, and high adhesion to supports [1]. These properties, combined with thermal stability, excellent corrosion resistance, low hysteresis, high sensitivity, and high reversible redox reaction, make Ruthenium oxide ($RuO_2$) suitable for a variety of applications in either a crystalline or hydrated amorphous state.

Over time, increasing interest in many areas of research has been reported. In the literature, we can find applications for $CO_2$ reduction [2], pH sensing [3], direct analysis and monitoring of engine lubricating oil conditions [4], and water splitting [5]. Nonetheless, due to the high value of electrochemical capacitance, voltage range, reversibility, and high charge/discharge capability, a large amount of research currently focuses on applications for electrochemical energy storage (EES) systems, as anodes for lithium battery [6,7] or, mostly, as supercapacitors [8,9].

Despite the obtained result, there remain some issues to cope with in regard to practical application and large-scale production.

As ruthenium is a noble, and, therefore, expensive metal, one of the main objectives is currently aimed at reducing the loading of this metal. To cope with this issue, dilution strategies with non-noble metals, such as manganese [10–12], as well as use as nanoparticle coatings of low-cost materials [13,14] and thin films [8,15,16], have been proposed. The latter presents the additional advantage of making the oxide structure less resistant and more capacitive [17].

Thin films of ruthenium oxide have been grown by adopting different techniques, such as reactive sputtering [18], electrodeposition [19] (even under pulsed modality [20,21]), electro-spray deposition [22], as well as powder-to-electrode [23]. Other common routes include thermal decomposition of sol-gel [24,25] of hydroalcoholic solutions containing salt precursors [1,26] where the surface coating is generally conducted by drop or dip casting.

Since innovative methods are often expensive, while dip or drop cast results in thick and uncontrolled coatings, a technique that is increasingly adopted to obtain the deposition of thin and highly reproducible films on flat surfaces is spin coating [27,28]. Spin coating implies different steps. First, the solution containing the precursor salt is dispensed in the center of a high-velocity rotating substrate. Then, due to the centrifugal force, the solution is spread radially. After deposition, the solvent is evaporated. The final characteristics of the film will depend on the nature of the solution and the selected spin process parameters.

Previous papers studied the effect of spin coating conditions mainly on morphological [29,30] or optical property aspects of the prepared films [30,31], while few papers have investigated the electrochemical response [25].

The present work is aimed at preparing spin-coated electrodes consisting of a Ruthenium oxide film, as a model coating, and studying the influence of some parameters related to the use of spin coating on the morphology and electrochemical performance of the material.

In particular, 10 different electrodes were fabricated and compared, varying both parameters that affect the thickness of the film (number of depositions and speed) and parameters that affect its uniformity and density (volume and concentration of the deposited material). The performance of thin films was evaluated by morphological analysis by scanning electron microscopy (SEM) coupled with energy-dispersive X-ray spectroscopy (EDX), cyclic voltammetry in the presence and absence of chlorides, and linear sweep voltammetry.

## 2. Materials and Methods

### 2.1. Materials

All the reagents, including Titanium foils (127 μm thickness, 99.6% purity), were provided by Sigma-Aldrich (Milano, Italy).

### 2.2. Electrode Preparation

Rectangular samples of the titanium foil were cut into $1.5 \times 2$ cm$^2$ pieces. All samples were cleaned in an ultrasonic bath first with water/acetone solution (*v/v*, 50/50) for 10 min, then another 10 min in ethanol, and finally dried under a nitrogen stream at ambient temperature.

Each of the reported deposits was performed in duplicate to ensure the reproducibility of the method and for greater accuracy in data collection.

All the samples (except E-10, see Table 1) were spin coated (SC) using an alcoholic solution containing RuCl$_3$ as the precursor salt. A 20 μL volume was used for each deposit, except for sample E-01, for which only 10 μL of precursor salt solution was used. The duration of the deposition was kept constant in all the tests and was equal to 30 s. This value is the result of a preliminary optimization.

The sample E-10 was obtained by drop-casting (DC) the same precursor solution on a single face.

**Table 1.** List of fabricated electrodes.

| Id | Method | N. Deposits | Volume (µL) | Spin Speed (min$^{-1}$) | [RuCl$_3$] (mol·L$^{-1}$) | T (°C) |
|----|--------|-------------|-------------|-------------------------|---------------------------|--------|
| E-01 | SC | 3 | 10 | 500 | 0.1 | 400 |
| E-02 | SC | 3 | 20 | 500 | 0.1 | 400 |
| E-03 | SC | 3 | 20 | 1000 | 0.1 | 400 |
| E-04 | SC | 3 | 20 | 250 | 0.1 | 400 |
| E-05 | SC | 1 | 20 | 500 | 0.1 | 400 |
| E-06 | SC | 2 | 20 | 500 | 0.1 | 400 |
| E-07 | SC | 6 | 20 | 500 | 0.1 | 400 |
| E-08 | SC | 3 | 20 | 500 | 0.05 | 400 |
| E-09 | SC | 3 | 20 | 500 | 0.01 | 400 |
| E-10 | DC | 3 | 20 | 500 | 0.1 | 400 |
| E-11 | SC | 3 | 20 | 500 | 0.1 | 100 [a] |

[a] temperature adopted during the intermediate steps.

Samples were then calcinated in a muffle furnace for 10 min at 400 °C (except E-11, for which a temperature of 100 °C was adopted during the intermediate steps). The treatment was repeated a certain number of times, as specified in Table 1, with final calcination for 1 h at 400 °C.

During electrochemical tests, the electrode was covered by an insulating tape to obtain a total exposed surface of 1 cm$^2$.

Several electrodes were prepared (Table 1) by varying the spin coating rate in the range 250–1000 rpm, the number of deposits in the range 1–6, and the concentration of RuCl$_3$ solution in the range 0.01–0.1 mol·L$^{-1}$.

*2.3. Electrode Characterization*

The surface morphologies of the electrodes were examined with a scanning electron microscope (SEM, Zeiss Gemini SIGMA 300 FEG SEM (Jena, Germany)), equipped with an energy-dispersive X-ray spectroscopy (EDX) detector (Bruker).

The atomic ratio (AR) of Ruthenium to the total metal content (Ruthenium + Titanium) was determined as the average value detected on a central area of $100 \times 100$ mm$^2$, according to the Equation (1):

$$[AR] = \frac{\%_{At}(Ru)}{\%_{At}(Ru) + \%_{At}(Ti)} \tag{1}$$

Electrochemical measurements were performed with an Amel System 5000 in a three electrode cell. Acquisition software was a CorrWare version 3.5c Scribner; elaboration software was a CorrView version 3.5c Scribner. The prepared electrodes were employed as working electrodes (exposed surface of 1 cm$^2$) using a platinum counter electrode and a saturated Ag/AgCl reference electrode.

Cyclic and linear sweep voltammograms (CVs and LSVs) were recorded, at ambient temperature (22 ± 1 °C), in an aqueous solution of Na$_2$SO$_4$ (0.1 mol·L$^{-1}$) at scan rates of 50, 100, and 200 mV·s$^{-1}$. To quantitatively evaluate the charge storage capacity, the positive voltametric charge ($q_+^*$) was determined by graphical integration of the voltammograms recorded at 50 mV·s$^{-1}$ in the potential window of 0–1 V [32].

$$q_+^* = \frac{1}{v} \int_{0\,V}^{1V} i\,dV \tag{2}$$

CVs and LSVs of the Fe(CN)$_6{}^{3-/4-}$ redox couple were recorded in an aqueous solution of Na$_2$SO$_4$ (0.1 mol·L$^{-1}$) containing 5 mM of K$_3$Fe(CN)$_6$ and 5 mM of K$_4$Fe(CN)$_6$ at the same scan rates. CVs and LSVs of the solution containing chloride were recorded in an aqueous solution of NaCl (0.1 mol·L$^{-1}$) at the same scan rates.

## 3. Results and Discussion

The prepared electrodes and their preparation conditions are shown in Table 1.

The morphology, surface composition, and electrochemical behavior of the electrodes were analyzed in detail.

### 3.1. Morphological Characterization

The samples were first subjected to morphological characterization by SEM-EDX analysis. The images shown in Figure 1 refer to only E-02, E-07, E-10, and E-11 to compare the sample E-02, considered as a referent, with a sample presenting an increased number of deposits (E-07), with a sample obtained via drop cast (E-10), and with a sample subjected to a milder intermediate thermal treatment (E-11), respectively. The deposits displayed a substantially even distribution of the oxide film with the presence of some RuOx aggregates probably due to insufficient mixing of the solution or to imperfections of the surface. This result was also confirmed by the element distribution maps, here included as insets.

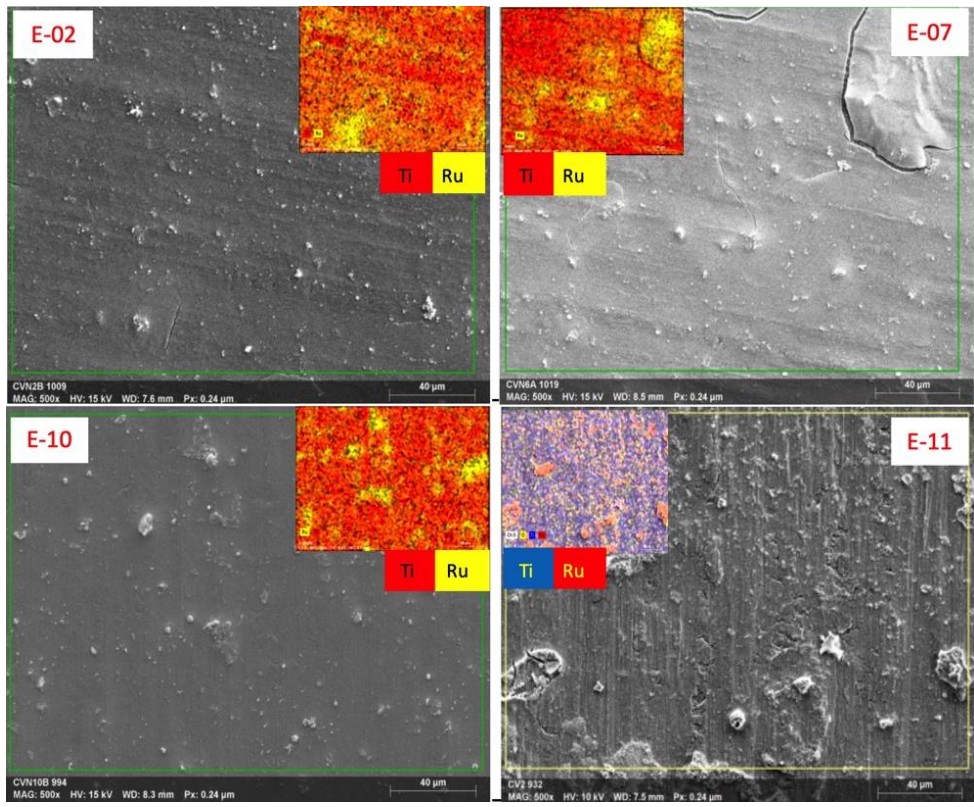

**Figure 1.** FE-SEM for some selected electrodes. Insets are the corresponding energy-dispersive X-ray spectroscopy (EDX) map.

Some exfoliated areas were detected near the edges probably where the film was thinner due to the radial distribution of the deposit on a rectangular substrate. Only in the thickest sample the deposit took on mud-cracked structures typical of thick (more massive) ruthenium oxide films.

The greater inhomogeneity of sample E-11 can be attributed to the adopted thermal treatment. In fact, the application of intermediate low-temperature treatments favors the evaporation of the solvent without converting the ruthenium salt into oxide, thus promoting the formation of larger aggregates with poor adhesion.

From EDX data, for all the samples, the atomic ratio of ruthenium to the amount of ruthenium and titanium was calculated using Equation (1) (Figure 2a). This value provides a rough estimate of the Ruthenium present on the electrode surface. The effects of the

concentration of the precursor salt (Figure 2b), the number of deposits (Figure 2c), and the rotation speed (Figure 2d) were also reported separately.

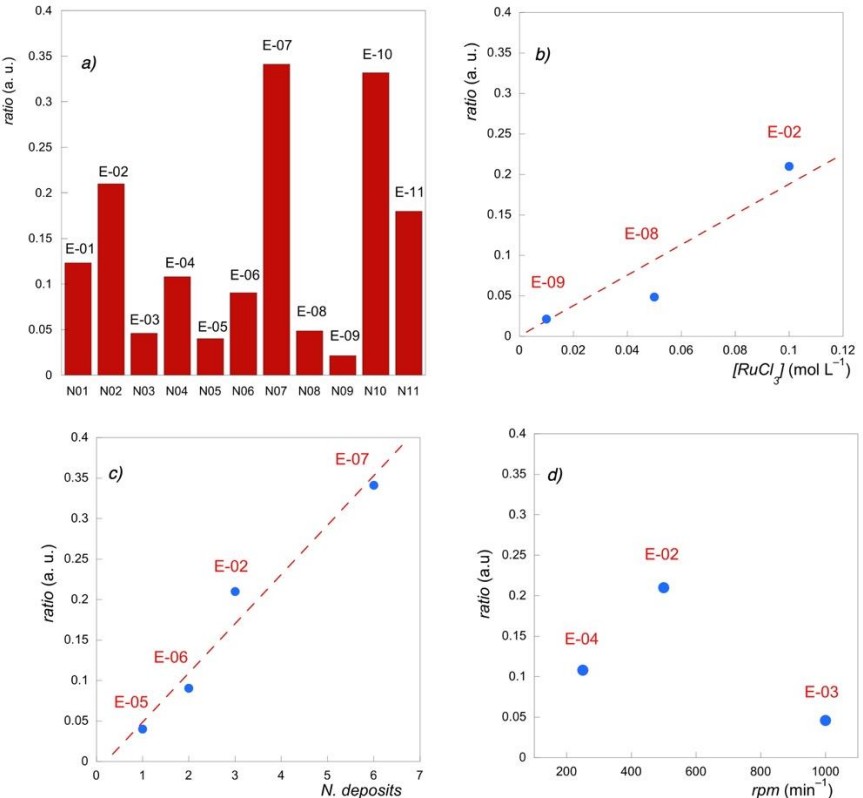

**Figure 2.** Atomic ratio of Ruthenium to the total metal content from EDX values (**a**) for all the electrodes; (**b**) vs. precursor concentration; (**c**) vs. number of deposits; (**d**) vs. spin rate.

In general, the Ru abundance on the electrode surface increased when a larger amount of precursor was provided. Therefore, the samples with the highest Ru atomic ratio were those prepared with a greater number of deposits (E-07), which were prepared by drop cast (E-10) and with a higher volume (E-02 > E-01). In particular, a linear dependence with the number of deposits (E-07 > E-02 > E-06 > E-05) and the concentration of the precursor salt (E-02 > E-08 > E-09) was found.

Instead, the variation of the Ru atomic ratio with the spin coater speed presented a maximum corresponding to 500 rpm (E-02), since higher speed (E-03) contributed to wiping out the solution, while slow rotations (E-04) did not favor its homogeneous dispersion.

### 3.2. Cyclic Voltammetry

The capacitive behavior of the films was investigated by cyclic voltammograms recorded in $Na_2SO_4$ 0.05 mol·L$^{-1}$, in the potential range 0.0–1.0 V, thus excluding the faradic peak portion, at a scan rate of 50 mV·s$^{-1}$.

This test provides information on the amount of charge transferred during the electrochemical reaction.

The voltametric charge of the positive sweep, $q_+^*$ reported in Table 2, is mainly determined by the proton mobility in solution and by the electron transfer rate [32]. In this case, by adopting circumneutral electrolyte conditions, the contribution of the solution is negligible, and the accumulation of charge is determined by the transition to higher oxidation states of the surface oxide layer. By minimizing the contribution of the ionic conductivity of the solution, it is possible to better evaluate the intrinsic properties of the material.

**Table 2.** Electrochemical data.

| Id | $q_+^*$ (mC cm$^{-2}$) | $\Delta E$ [a] (Volt) | $I_{p(c)}/I_{p(a)}$ [b] | E @ 3 mA·cm$^{-2}$ [c] (Volt) |
|---|---|---|---|---|
| E-01 | 3.30 | 0.174 | 0.96 | 1.426 |
| E-02 | 5.84 | 0.182 | 0.91 | 1.276 |
| E-03 | 2.98 | 0.193 | 0.94 | 1.402 |
| E-04 | 3.18 | 0.194 | 0.93 | 1.404 |
| E-05 | 3.70 | 0.210 | 0.90 | 1.381 |
| E-06 | 6.50 | 0.212 | 0.92 | 1.361 |
| E-07 | 5.76 | 0.149 | 0.97 | 1.229 |
| E-08 | 6.08 | 0.179 | 0.96 | 1.287 |
| E-09 | 3.44 | 0.146 | 0.93 | 1.303 |
| E-10 | 8.40 | 0.230 | 0.95 | 1.334 |
| E-11 | 8.88 | 0.213 | 0.86 | 1.268 |

[a] $(E_a - E_c)$ from cyclic voltammetry at 50 mV·s$^{-1}$ in presence of the Fe(CN)$_6^{3-/4-}$ redox couple. [b] Peak current ratio from *CVs* at 50 mV·s$^{-1}$ in presence of the Fe(CN)$_6^{3-/4-}$ redox couple. [c] Potential corresponding at 3 mA·cm$^{-2}$ current density in 0.1 M Na$_2$SO$_4$ at 50 mV·s$^{-1}$

As can be seen in Figure 3, the voltametric charge was only slightly influenced by the concentration of ruthenium precursor salt, while a significant positive influence of the number of deposits was found.

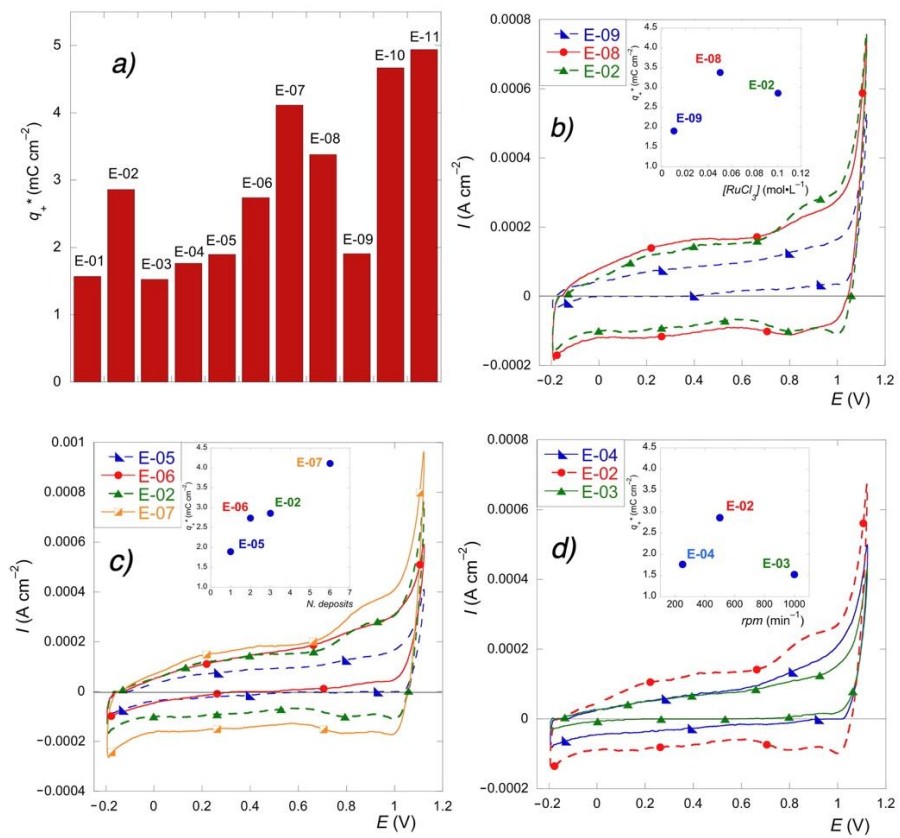

**Figure 3.** (**a**) Voltametric charge ($q_+^*$) for all the electrodes. Effect of precursor concentration (**b**) number of deposits (**c**) and spin rate (**d**) on cyclic voltammograms and voltametric charge (inset).

The samples with the highest voltametric charge were those deposited by drop cast (E-10), those treated with low-temperature intermediate steps (E-11), and those obtained by high-cycle deposition (E-07), thus indicating that the charge increased with the oxide loading. This is generally explained [33] considering that high amounts of deposited material imply a high surface concentration of active sites.

This was also further confirmed by the effect of speed rotation. In fact, the intermediate speed value (E-02), which enabled the deposition of the most abundant amount of precursor solution, under the adopted conditions, presented a broader voltametric area than the other two investigated values (E-03 and E-04).

### 3.3. Cyclic Voltammetry in the Presence of a Redox Couple

Ruthenium oxide films exhibit a well-known behavior as metal conductors [1]. To evaluate the faradic properties of the electrodes, and in particular to highlight the differences in the electron transfer, cyclic voltammetry of the ferrocyanide/ferricyanide redox couple was carried out at different scan rates.

It is widely accepted that an electrochemically reversible process occurs when there is no major structural rearrangement of the redox couple or when the electron transfer rate is higher than the mass transport rate [33]. The higher the discrepancy from the reversible behavior, the slower the electron transfer of the electrode, all other parameters being constant.

Data collected in Table 2 show that the current of the reverse peak to that of the forward peak ratio, $i_p/i_{pf}$, was approximately 1 for all the investigated samples, thus indicating a high reversible behavior of the redox couple with the prepared films [33].

Only the electrode E-11 presented a lower value.

The film showing the greatest reversibility is that prepared with the highest number of deposits (E-07), while the worst reversibility is displayed by films prepared with low-temperature intermediate steps where the deposit inhomogeneity probably slows down the electron transfer. This result is further confirmed by measurements of anodic-to-cathodic potential separation using Equation (3), which, in a reversible system, is at 25 °C,

$$Ep(a) - Ep(c) = \Delta Ep = 59 \, (mV)/n \tag{3}$$

where n corresponds to the number of exchanged electrons.

Data in Table 2 show that for slow transfers of electrons to the electrode surface, i.e., for more irreversible processes, the difference of the peak potentials increases. However, limited deviations from the theoretical value, especially in voltammetry conducted at high scan rates, are still indicative of reversible systems.

Figure 4 shows the effect of the operative conditions on the separation potential value.

Reversibility was almost unaffected by both the concentration of the precursor salt and spin rate. Again, the most influential parameter was the number of deposits. The higher the number of deposits and, therefore, the film thickness, the higher the electron transfer rates in the electrochemical reaction of the considered mediators.

### 3.4. Linear Sweep Voltammetry

Another interesting aspect to investigate the electrochemical behavior of the materials is the potential at which the Oxygen evolution reaction occurs. RuOx- based electrodes, like all active anodes (e.g., graphite, $IrO_2$, and Pt electrodes), imply chemisorption of hydroxyl radicals and present potential for $O_2$ evolution lower than 1.8 V/SHE [34].

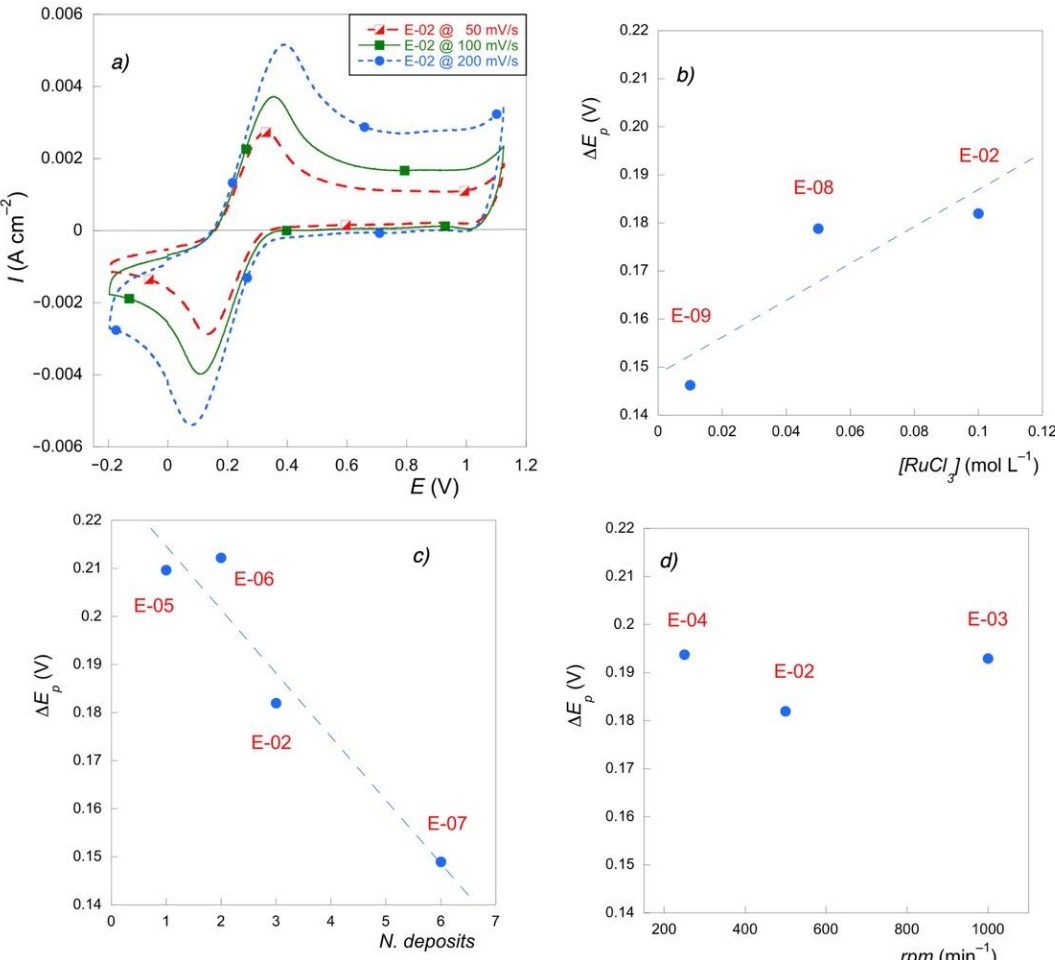

**Figure 4.** (**a**) Cyclic voltammetry for E-02 at different rates in presence of Fe(CN)$_6$$^{3-/4-}$ couple; (**b**) $\Delta E_p$ (V) vs. precursor concentration; (**c**) $\Delta E_p$ (V) vs. number of deposits; (**d**) $\Delta E_p$ (V) vs. spin rate.

Linear sweep voltammetry was recorded in 0.1 mol·L$^{-1}$ Na$_2$SO$_4$ solution in the potential range 0.4–1.6 V and with the scan rate of 50 mV·s$^{-1}$.

The activity of the electrodes, as shown in Figure 5a, differed significantly.

By comparing, for example, the potential values corresponding to the current density value of 0.003 A·cm$^{-2}$, we can notice that the thermal treatments almost did not affect the catalytic properties, which could be significantly enhanced by increasing the number of depositions (E-07). In this last case, the peak shifted to a lower potential, thus indicating that the active sites promoting oxygen evolution reaction (OER) are more easily activated. The electrode prepared with the lower volume of precursor salt solution showed the worst performance with the highest total overpotential for OER. Poor performance at high current densities was also exhibited by the drop cast electrode, which was characterized by the highest voltametric charge (E-10). This behavior is not unusual and has also been reported in previous papers [33]. The inversion of performance with an increasing current for E-02 and E-11 could be attributed to the resistance of the catalyst layer [35].

Figure 5b–d reveal that electrocatalytic activity for OER is mostly related to the number of depositions and only slightly affected by the concentration of the precursor salt. A beneficial effect was shown by the intermediate value of rotation speed since high rotation speed results in thinner deposits, while a low centrifugal force does not effectively contrast the liquid surface energy, thus resulting in an uneven layer.

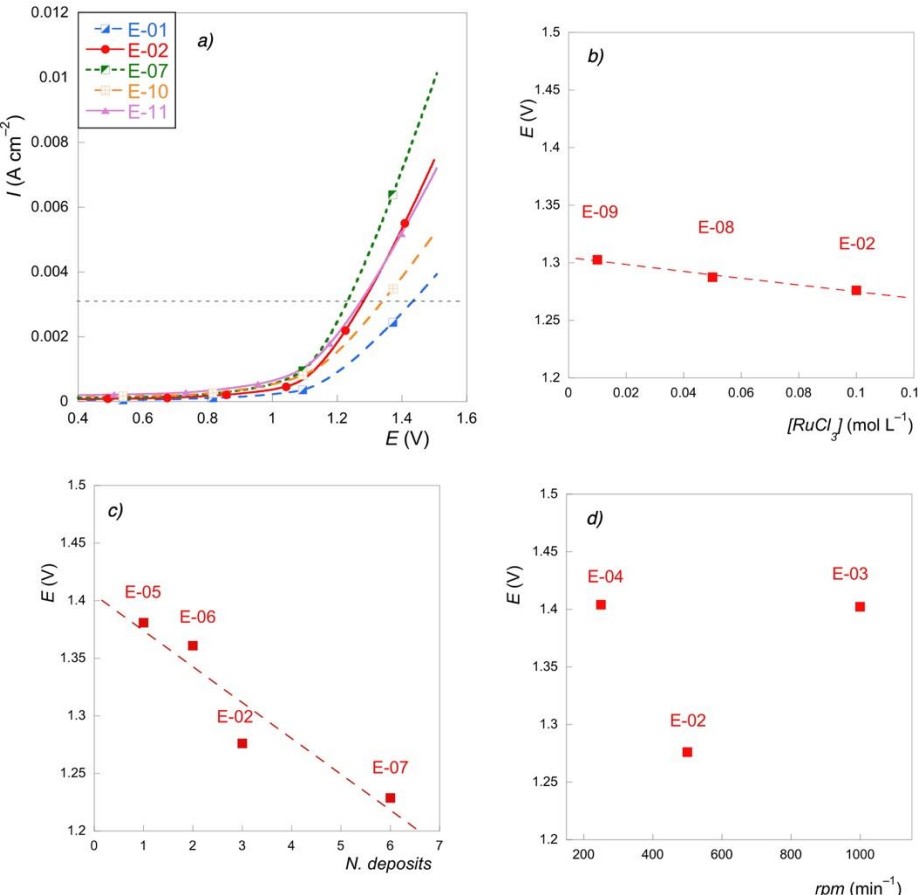

**Figure 5.** (**a**) LSV for some selected electrodes; (**b**) E(V) @ 3 mA·cm$^{-1}$ vs. precursor concentration; (**c**) E(V) @ 3 mA·cm$^{-1}$ vs. number of deposits; (**d**) E(V) @ 3 mA·cm$^{-1}$ vs. spin rate.

*3.5. Cyclic Voltammetry with Chlorides*

Finally, the cyclic voltammetry was repeated by replacing the supporting electrolyte with 0.1 M NaCl.

These measurements can highlight the tendency of the material to oxidize the chloride, thus producing intermediates, such as active chlorine, with implications in the indirect oxidation or disinfection processes. In Figure 6a (inset), the voltammograms obtained with and without chlorides in the potential range from −0.2 to +1.12 V for electrode E-02 are reported. In this range, the oxidation of the solvent/electrolyte has not yet started. The two cycles substantially overlap, i.e., the change in electrolyte did not change the capacitive response of all electrodes, also confirming that the pseudocapacitive behavior under the adopted conditions (circumneutral pH) is not due to the ionic mobility in solution but to the electron properties of the Ru films. This behavior is quite similar for all electrodes tested. However, at higher potential values, the cycles differed significantly (Figure 6a). A slight shift of the solvent discharge towards less positive values can be found, together with a typical partially reversible wave at about 1.1 V, generally attributed to the reduction of oxidized chlorine-related species [36].

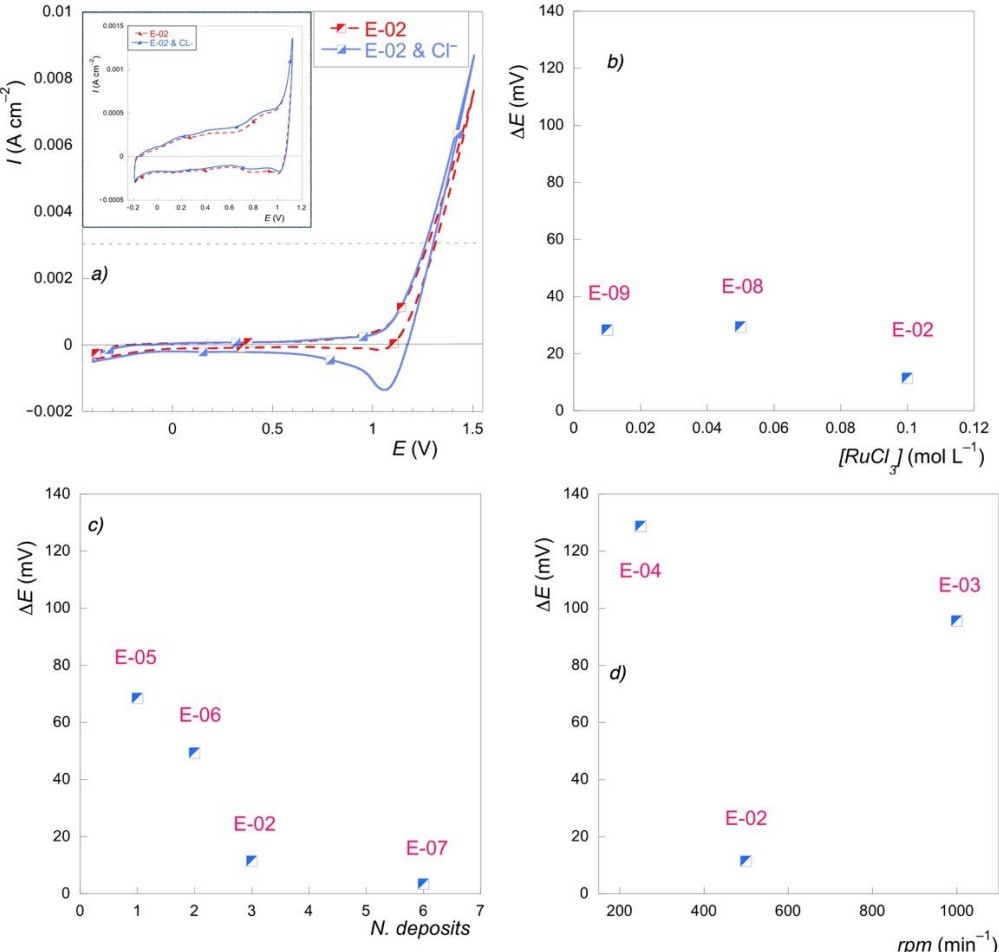

**Figure 6.** (**a**) Cyclic voltammetry for E-02 with $Na_2SO_4$ 0.1 M (dot line) and with NaCl 0.1 M (solid line); (**b**) [$\Delta E(V) = E_{(Na2SO4)} - E_{(NaCl)}$] @ 3 mA·cm$^{-1}$ vs. precursor concentration; (**c**) [$\Delta E(V) = E_{(Na2SO4)} - E_{(NaCl)}$] @ 3 mA·cm$^{-1}$ vs. number of deposits; (**d**) [$\Delta E(V) = E_{(Na2SO4)} - E_{(NaCl)}$] @ 3 mA·cm$^{-1}$ vs. spin rate.

The comparison of the potentials corresponding to the current of 3 mA·cm$^{-2}$, obtained from the voltammetry recorded with and without chlorides, indicates that the discrepancy, and, therefore, the selectivity, remains constant as the concentration of the precursor salt solution varies. This can be explained assuming that, in the considered range, even the lowest tested concentration provides a consistent amount of RuOx (Figure 6b). Instead, the difference between the potentials tends to decrease as the number of deposits increases (Figure 6c) and with the central rotation speed value (Figure 6d), which has been, to date, attributed to the greater amount of deposit and, therefore, to the greater concentration of active sites.

## 4. Conclusions

Spin coated thin films of Ruthenium oxide were prepared, and several parameters influencing the deposition were evaluated. The results indicate that the operative conditions affect the morphology and electrochemical response of the films, thus making them suitable for different specific applications.

To enhance both the capacitive properties and the electrocatalytic activity for the oxygen evolution reaction, it is essential to increase the deposit load (acting on the number of cycles and less significantly on the concentration of the precursor salt).

To promote the selectivity towards the chlorine oxidation reaction with respect to oxygen evolution, it is advisable to prepare films that contain a more limited amount of

ruthenium. This also corroborates the adoption of strategies aimed at diluting ruthenium with additional metals to mitigate the production of undesired chlorinated intermediates.

Finally, in the investigated range of values and considering the viscosity of the solution, the intermediate value of the spin rate (corresponding to 500 rpm) represents the best compromise between the excessive dispersion, obtained at a higher speed, and the incomplete distribution, observed at a lower speed.

**Author Contributions:** E.P., G.S. and S.D.S. prepared the samples and performed the physical and morphological analysis. E.P., G.S., S.D.S. and M.O., F.P. provided supervision. E.P. and G.S. wrote the manuscript. All authors have read and agreed to the published version of the manuscript.

**Funding:** This research received no external funding.

**Institutional Review Board Statement:** Not applicable.

**Informed Consent Statement:** Not applicable.

**Data Availability Statement:** Not applicable.

**Conflicts of Interest:** The authors declare no conflict of interest.

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
