# Peer review of "Effect of Spin Coating Parameters on the Electrochemical Properties of Ruthenium Oxide Thin Films"

_2673-3293, doi:10.3390/electrochem2010008_

Round 1
Reviewer 1 Report
Review of electrochem-1100406
The manuscript by Petrucci and co-workers presents a reminder that the conditions of preparation of an electrode influences the properties of the electrode. Here, RuCl3 was spin-coated onto Ti-foils to investigate how the spin-coating conditions affect the resulting electrodes.
The results are more or less expected – more material means better electrodes. The problem with the manuscript is that the authors only present results from a single electrode of each type. To be able to draw any firm conditions, several electrodes should have been prepared from each recipe. We can take the SEM section as an example: This section needs a measure of variation. The inhomogeneity of the films, as can be seen from the EDX, should be quantified in a consistent manner, maybe the ratio can be measured at 10 smaller areas on each film. But the variation at repeated sample preparations also needs to be quantified. n=1 is not really good enough. I know EXX mapping is extremely time-consuming, but much of this could have been done by pointwise sampling.
We are also told nothing about the surface coverage. Do the authors have any estimate of how much of the surface is covered by the RuOx? It’s really hard to see from the EDX maps, since the Ru gets overwhelmed by the colour of the Ti signal. Is it a good coverage or are large areas pure titanium (oxide, I guess after the heat treatment)? How thick are the deposits?
I am quite surprised that the charge-transfer and capacitive measurements were performed using only voltammetry. Electrochemical impedance spectroscopy seems to have been a relevant tool here. I would also suggest that the authors show the data for the clean Ti electrodes as a comparison.
All the SEM and EDX data and all the electrochemical data for all electrodes should be included as supplementary information for the interested reader.
The work presented here is interesting, but the lack of many relevant points, and crucially, the lack of repeated measurements diminishes the value of the article. I would suggest that the authors present data for at least a few samples of each kind, just to make sure we know what the variation within in measurement to be able to draw accurate conclusions.
Reviewer 2 Report
In their manuscript, Dr Petrucci et al. describe the fabrication of Ruthenium oxide films based on spin coating technique with subsequent annealing at 400°C. They studied the influence of different spin coating parameters such as spinning speed, volume, concentration. The morphology and electrochemical properties were investigated to determine the most relevant parameters. The study is quite interesting and the manuscript is well-written. Thus, I recommended this manuscript for publication in Electrochem once the authors revise it by addressing the following minor points:
- In my opinion the Figure 3 should be modify. The CVs are not clear (visible). I think it will be better if the CVs are the main plots and charge vs factors are in inset.
- The figure 6.a is not clear for me. There are two CVs but I do not understand the difference. Do not forget to mention the figure 6.b-d in the main text.
Round 2
Reviewer 1 Report
The authors have adressed my questions, and I have no further comments. The paper can be published in its present form.